# Learning Invariant Molecular Representation in Latent Discrete Space

**Xiang Zhuang**[1,2,3*], **Qiang Zhang**[1,2,3*†]**, Keyan Ding**[2]**, Yatao Bian**[4]**,**
**Xiao Wang**[5]**, Jingsong Lv**[6]**, Hongyang Chen**[6]**, Huajun Chen**[1,2,3†]
[1]College of Computer Science and Technology, Zhejiang University
[2]ZJU-Hangzhou Global Scientific and Technological Innovation Center
[3]Zhejiang University - Ant Group Joint Laboratory of Knowledge Graph
[4]Tencent AI Lab, [5]School of Software, Beihang University, [6]Zhejiang Lab
`{zhuangxiang,qiang.zhang.cs,dingkeyan,huajunsir}@zju.edu.cn`
`yatao.bian@gmail.com,xiao_wang@buaa.edu.cn`
`{jingsonglv,hongyang}@zhejianglab.com`

## Abstract

Molecular representation learning lays the foundation for drug discovery. However, existing methods suffer from poor out-of-distribution (OOD) generalization, particularly when data for training and testing originate from different environments. To address this issue, we propose a new framework for learning molecular representations that exhibit invariance and robustness against distribution shifts. Specifically, we propose a strategy called "first-encoding-then-separation" to identify invariant molecule features in the latent space, which deviates from conventional practices. Prior to the separation step, we introduce a residual vector quantization module that mitigates the over-fitting to training data distributions while preserving the expressivity of encoders. Furthermore, we design a task-agnostic self-supervised learning objective to encourage precise invariance identification, which enables our method widely applicable to a variety of tasks, such as regression and multi-label classification. Extensive experiments on 18 real-world molecular datasets demonstrate that our model achieves stronger generalization against state-of-the-art baselines in the presence of various distribution shifts. Our code is available at `https://github.com/HICAI-ZJU/iMoLD`.

## 1 Introduction

Computer-aided drug discovery has played an important role in facilitating molecular design, aiming to reduce costs and alleviate the high risk of experimental failure [1, 2]. In recent years, the emergence of deep learning has led to a growing interest in molecular representation learning, which aims to encode molecules as low-dimensional and dense vectors [3, 4, 5, 6, 7, 8]. These learned representations have demonstrated their availability in various tasks, including target structure prediction [9], binding affinity analysis [10], drug re-purposing [11] and retrosynthesis [12].

Despite the significant progress in molecular representation methods, a prevailing assumption in the traditional approaches is that data sources are independent and sampled from the same distribution. However, in practical drug development, molecules exhibit diverse characteristics and may originate from different distributions [13, 14]. For example, in virtual screening scenarios [15], the distribution shift occurs not only in the molecule itself, e.g., size [13] or scaffold [16] changes, but also in the

---

*Equal contribution.
†Corresponding author.

target [14], e.g., the emergent COVID-19 leads to a new target from unknown distributions. This out-of-distribution (OOD) problem poses a challenge to the generalization capability of molecular representation methods and results in the degradation of performance in downstream tasks [17, 18].

Current studies mainly focus on regular Euclidean data for OOD generalization. Most studies [18, 19, 20, 21, 22] adopt the invariance principle [23, 24], which highlights the importance of focusing on the critical causal factors that remain invariant to distribution shifts while overlooking the spurious parts [23, 19, 20]. Although the invariance principle has shown effectiveness on Euclidean data, its application to non-Euclidean data necessitates further investigation and exploration. Molecules are often represented as graphs, a typical non-Euclidean data, where atoms as nodes and bonds as edges, thereby preserving rich structural information [3]. The complicated molecular graph structure makes it challenging to accurately distinguish the invariant causal parts from diverse spurious correlations [25].

Preliminary studies have made some attempts on molecular graphs [26, 25, 27, 28, 29]. They explicitly divide graphs to extract invariant substructures, at the granularity of nodes [28], edges [26, 27, 28, 29], or motifs [25]. These attempts can be summarized as the "first-separation-then-encoding" paradigm (Figure 1 (a)), which first divides the graph into invariant and spurious parts and then encodes each part separately. We argue that this practice is suboptimal for extremely complex and entangled graphs, such as real-world molecules [30, 31], since some intricate properties cannot be readily determined by analyzing a subset of the molecular structure [30, 31]. Besides, some methods [25, 27] require assumptions and inferences about the environmental distribution, which are often untenable for molecules due to the intricate environment. Additionally, the downstream tasks related to molecules are diverse, including regression and classification. However, some methods such as CIGA [26] and DisC [32] can only be applied to single-label classification tasks, due to the constraints imposed by the invariant learning objective function.

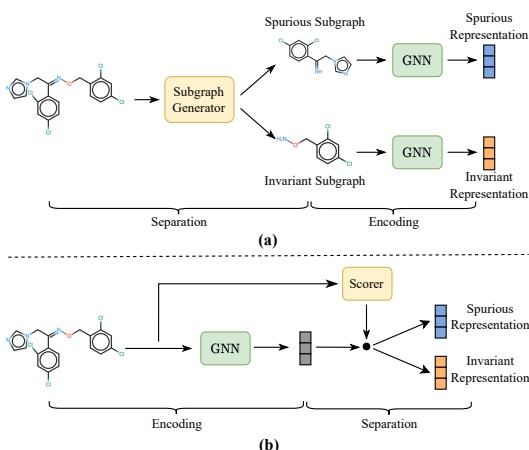

Figure 1: (a) First-Separation-Then-Encoding: the input is separated using a subgraph generator, then each subgraph is encoded respectively. (b): First-Encoding-Then-Separation: the input is encoded, then the representation is separated by a scorer.

To fill these gaps, we present a novel molecule invariant learning framework to effectively capture the invariance of molecular graphs and achieve generalized representation against distribution shifts. In contrast to the conventional approaches, we propose a "first-encoding-then-separation strategy" (Figure 1 (b)). Specifically, we first employ a Graph Neural Network (GNN) [33, 34, 35] to encode the molecule (i.e., encoding GNN), followed by a residual vector quantization module to alleviate the over-fitting to training data distributions while preserving the expressivity of the encoder. We then utilize another GNN to score the molecule representation (i.e., scoring GNN), which measures the contributions of each dimension to the target in latent space, resulting in a clear separation between invariant and spurious representations. Finally, we design a self-supervised learning objective [36, 37, 38] that aims to encourage the identified invariant features to effectively preserve label-related information while discarding environment-related information. It is deserving of note that the objective is task-agnostic, which means that our method can be applied to various tasks, including regression and single- or multi-label classification.

Our main contributions can be summarized as follows:

- We propose a paradigm of "first-encoding-then-separation" using an encoding GNN and a scoring GNN, which enables us to effectively identify invariant features from highly complex graphs.

- We introduce a residual vector quantization module that strikes a balance between model expressivity and generalization. The quantization acts as the bottleneck to enhance generalization, while the residual connection complements the model's expressivity.

- We design a self-supervised invariant learning objective that facilitates the precise capture of invariant features. This objective is versatile, task-agnostic, and applicable to a variety of tasks.

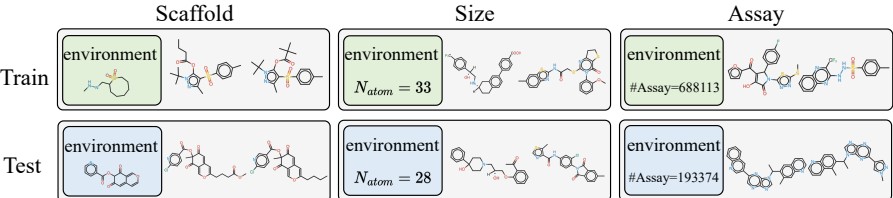

Figure 2: An overview of distribution shifts in molecules. Distribution shifts occur when molecules originate from different scaffold, size or assay environments.

- We conduct comprehensive experiments on a diverse set of real-world datasets with various distribution shifts. The experimental results demonstrate the superiority of our method compared to state-of-the-art approaches.

## 2 Related Work

**OOD Generalization and Invariance Principle.** The susceptibility of deep neural networks to substantial performance degradation under distribution shifts has led to a proliferation of research focused on out-of-distribution (OOD) generalization [39]. Three lines of methods have been studied for OOD generalization on Euclidean data, including group distributionally robust optimization [40, 41, 42], domain adaptation [43, 44, 45] and invariant learning [19, 20, 21]. Group distributionally robust optimization considers groups of distributions and optimize across all groups simultaneously. Domain adaptation aims to align the data distributions but may fail to find an optimal predictor without additional assumptions [19, 20, 46]. Invariant learning aims to learn invariant representation satisfying the invariant principle [20, 24], which includes two assumptions: (1) sufficiency, meaning the representation has sufficient predictive abilities, (2) invariance, meaning representation is invariant to environmental changes. However, most methods require environmental labels, which are expensive to obtain for molecules [26], and direct application of these methods to complicated non-Euclidean molecular structure does not yield promising results [13, 22, 26].

**OOD Generalization on Graphs.** Recently, there has been growing attention on the graph-level representations under distribution shifts from the perspective of invariant learning. Some methods [26, 32, 27, 28, 25] follows the "first-separation-then-encoding" paradigm, and they make attempt to capture the invariant substructures by dividing nodes and edges in the explicit structural space. However, these methods suffer from the difficulty in dividing molecule graphs in raw space due to the complex and entangled molecular structure [47]. Moreover, MoleOOD [25] and GIL [27] require inference of unavailable environmental labels, which entails prior assumptions on the environmental distribution. And the objective of invariant learning may hinder the application, e.g., CIGA [26] and DisC [32] can only apply to single-label classification rather than to regression and multi-label tasks. Additionally, OOD-GNN [48] does not use the invariance principle. It proposes to learn disentangled graph representations, but requires computing global weights for all data, leading to a high computational cost. OOD generalization on graphs can also be improved by another line of relevant works [29, 49, 47] on GNN explainability [50, 51], which aims to provide a rationale for prediction. However, they may fail in some distribution shift cases [26]. And DIR [29] and GSAT [49] also divide graphs in the raw structural space. Although GREA [47] learns in the latent feature space, it only conducts separation on each node while neglecting the different significance of each representation dimensionality. In this work, we focus on the OOD generalization of molecular graphs, against multi-type of distribution shifts, e.g., scaffold, size, and assay, as shown in Figure 2.

**Vector Quantization.** Vector Quantization (VQ) [52, 53] acts as a bottleneck of representation learning. It discretizes continuous input data in the hidden space by assigning them to the nearest vectors in a predefined codebook. Some studies [53, 54, 55, 56] have demonstrated its effectiveness to enhance model robustness against data corruptions. Other studies [57, 58, 59] find that taking VQ as an inter-component communication within neural networks can contribute to the model generalization ability. However, we posit that while VQ can improve generalization against distribution shifts, it may also limit the model's expressivity and potentially lead to under-fitting. To address this concern, we propose to equip the conventional VQ with a residual connection to strike a balance between model generalization and expressivity.

# 3 Preliminaries

## 3.1 Problem Definition

We focus on the OOD generalization of molecules. Let $\mathcal{G}$ be the molecule graph space and $\mathcal{Y}$ be the label space, the goal is to find a predictor $f : \mathcal{G} \to \mathcal{Y}$ to map the input $G \in \mathcal{G}$ into the label $Y \in \mathcal{Y}$. Generally, we are given a set of datasets collected from multiple environments $E_{all} : D = \{D^e\}_{e \in E_{all}}$. Each $D^e$ contains pairs of an input molecule and its label: $D^e = \{(G_i, Y_i)\}_{i=1}^{n_e}$ that are drawn from the joint distribution $P(G, Y|E = e)$ of environment $e$. However, the information of environment $e$ is always not available for molecules, thus we redefine the training joint distribution as $P_{train}(G, Y) = P(G, Y|E = e), \forall e \in E_{train}$, and the testing joint distribution as $P_{test}(G, Y) = P(G, Y|E = e), \forall e \in E_{all} \setminus E_{train}$, where $P_{train}(G, Y) \neq P_{test}(G, Y)$. We denote joint distribution across all environment as $P_{all}(G, Y) = P(G, Y|E = e), \forall e \in E_{all}$. Formally, the goal is to learn an optimal predictor $f^*$ based on training data and can generalize well across all distributions:

$$f^* = \arg\min_f \mathbb{E}_{(G, Y) \sim P_{all}} \left[ \ell \left( f \left( G \right), Y \right) \right], \tag{1}$$

where $\ell(\cdot, \cdot)$ is the empirical risk function. Moreover, since the joint distribution $P(G, Y)$ can be written as $P(Y|G)P(G)$, the OOD problem can be refined into two cases, namely covariate and concept shift [60, 61, 62, 63]. In covariate shift, the distribution of input differs. Formally, $P_{train}(G) \neq P_{test}(G)$ and $P_{train}(Y|G) = P_{test}(Y|G)$. While concept shift occurs when the conditional distribution changes as $P_{train}(Y|G) \neq P_{test}(Y|G)$ and $P_{train}(G) = P_{test}(G)$. We will consider and distinguish between both cases in our experiments.

## 3.2 Molecular Representation Learning

We denote a molecule graph by $G = \{\mathcal{V}, \mathcal{E}\}$, where $\mathcal{V}$ is the set of nodes (e.g., atoms) and $\mathcal{E} \in \mathcal{V} \times \mathcal{V}$ is the set of edges (e.g., chemical bonds). Generally, the predictor $f$ can be denoted as $\rho \circ g$, containing an encoder $g : \mathcal{G} \to \mathbb{R}^d$ that extracts representation for each molecule and a downstream classifier $\rho : \mathbb{R}^d \to \mathcal{Y}$ that predicts the label with the molecular representation. In particular, the encoder $g$ operates in two stages: firstly, by employing a graph neural network [33, 34, 35] to generate node representations $\mathbf{H}$ according to the following equation:

$$\mathbf{H} = \left[\mathbf{h}_1, \mathbf{h}_2, \ldots, \mathbf{h}_{|\mathcal{V}|}\right]^\top = \text{GNN}(G) \in \mathbb{R}^{|\mathcal{V}| \times d}, \tag{2}$$

where $\mathbf{h}_v \in \mathbb{R}^d$ is the representation of node $v$. Secondly, the encoder utilizes a readout operator to obtain the overall graph representation $\mathbf{z}$:

$$\mathbf{z} = \text{READOUT}(\mathbf{H}) \in \mathbb{R}^d. \tag{3}$$

The readout operator can be implemented using a simple, permutation invariant function such as average pooling.

# 4 Method

This section presents the details of our proposed method that learns **i**nvariant **Mo**lecular representation in **L**atent **D**iscrete space, called **iMoLD**. Figure 3 shows the overview of iMoLD, which mainly consists of three steps: 1) Using a GNN encoder and a residual vector quantization module to obtain the molecule representation (Section 4.1); 2) Separating the representation into invariant and spurious parts through a GNN scorer (Section 4.2); 3) Optimizing the above process with a task-agnostic self-supervised learning objective (Section 4.3).

## 4.1 Encoding with Residual Vector Quantization

The current mainstream methods [26, 25, 28, 27, 32] adopt the "first-separation-then-encoding" paradigm, which explicitly divides graphs into invariant and spurious substructures on the granularity of edge, node, or motif, and then encodes each substructure individually. In contrast, we use the opposite paradigm that first encodes the whole molecule followed by separation.

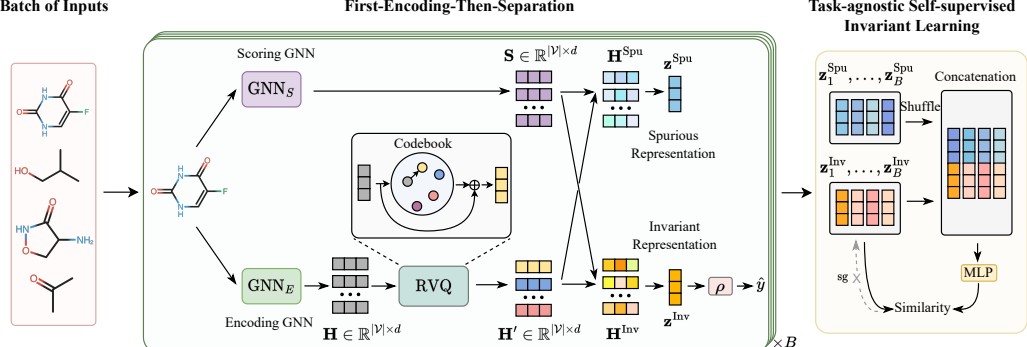

Figure 3: An overview of iMoLD. Firstly, given a batch of inputs, we learn invariant and spurious representations ($\mathbf{z}^{\text{Inv}}$ and $\mathbf{z}^{\text{Spu}}$) for each input in latent discrete space by a first-encoding-then-separation paradigm. An encoding GNN and an RVQ module are involved to obtain molecule representation, then the representation is separated through a scoring GNN. The invariant $\mathbf{z}^{\text{Inv}}$ is used to predict the label $\hat{y}$. Then a task-agnostic self-supervised learning objective across the batch is designed to facilitate the acquisition of reliable invariant $\mathbf{z}^{\text{Inv}}$.

Specifically, given an input molecule $G$, we first use a GNN to encode it, resulting in node representations $\mathbf{H}$:

$$\mathbf{H} = \text{GNN}_E(G) \in \mathbb{R}^{|\mathcal{V}| \times d}, \tag{4}$$

where $\text{GNN}_E$ represents the encoding GNN, $|\mathcal{V}|$ is the number of nodes in $G$, and $d$ is the dimensionality of features. Inspired by the studies [57, 59] that Vector Quantization (VQ) [52, 53] is helpful to improve the model generalization on computer vision tasks, we propose a Residual Vector Quantization (RVQ) module to refine the obtained representations.

In the RVQ module, VQ is used to discretize continuous representations into discrete ones. Formally, it introduces a shared learnable codebook as a discrete latent space: $\mathcal{C} = \{\mathbf{e}_1, \mathbf{e}_2, \ldots \mathbf{e}_{|\mathcal{C}|}\}$, where each $\mathbf{e}_k \in \mathbb{R}^d$. For each node representation $\mathbf{h}_v$ in $\mathbf{H}$, VQ looks up and fetches the nearest neighbor in the codebook and outputs it as the result. Mathematically,

$$Q(\mathbf{h}_v) = \mathbf{e}_k, \quad \text{where} \quad k = \underset{k \in \{1, \ldots, |\mathcal{C}|\}}{\text{argmin}} \|\mathbf{h}_v - \mathbf{e}_k\|_2, \tag{5}$$

and $Q(\cdot)$ denotes the discretization operation which quantizes $\mathbf{h}_v$ to $\mathbf{e}_k$ in the codebook.

The VQ operation acts as a bottleneck to enhance generalization and alleviate the easy-over-fitting issue caused by distribution shifts. However, it also impairs the expressivity of the model by using a limited discrete codebook to replace the original continuous input, suffering from a potential under-fitting issue. Accordingly, we propose to equip the conventional VQ with a residual connection to strike a balance between model generalization and expressivity. In specific, we incorporate both the continuous and discrete representations to update node representations $\mathbf{H}$ to $\mathbf{H}'$:

$$\mathbf{H}' = \left[ Q(\mathbf{h}_1) + \mathbf{h}_1, Q(\mathbf{h}_2) + \mathbf{h}_2, \ldots, Q(\mathbf{h}_{|\mathcal{V}|}) + \mathbf{h}_{|\mathcal{V}|} \right]^\top. \tag{6}$$

Similar to VQ-VAE [52, 53], we employ the exponential moving average updates for the codebook:

$$N_k^{(t)} = N_k^{(t-1)} * \eta + n_k^{(t)}(1 - \eta), \quad \mathbf{m}_k^{(t)} = \mathbf{m}_k^{(t-1)} * \eta + \sum_v^{n_k^{(t)}} \mathbf{h}_v^{(t)}(1 - \eta), \quad \mathbf{e}_k^{(t)} = \frac{\mathbf{m}_k^{(t)}}{N_k^{(t)}}, \tag{7}$$

where $n_k^{(t)}$ is the number of node representations in the $t$-th mini-batch that are quantized to $\mathbf{e}_k$, and $\eta$ is a decay parameter between 0 and 1.

## 4.2 Separation at Nodes and Features

After encoding, we separate the representation into invariant parts and spurious parts. It is worth noting that our separation is not only performed at the node dimension but also takes into account the feature dimension in the latent space. The reasons are two-fold: 1) Distribution shifts on molecules

can occur at both the structure level and the attribute level [26], corresponding to the node dimension $|\mathcal{V}|$ and the feature dimension $d$ in $\mathbf{H}'$, respectively. 2) The resulting representation may be highly entangled, thus it is advisable to perform a separation on each dimension in the latent space.

Specifically, we use another GNN as a scorer to obtain the separating score $\mathbf{S}$:

$$\mathbf{S} = \sigma\left(\text{GNN}_S\left(G\right)\right) \in \mathbb{R}^{|\mathcal{V}| \times d}, \tag{8}$$

where $\text{GNN}_S$ represents the scoring GNN, $|\mathcal{V}|$ is the number of nodes in $G$, and $d$ is the dimensionality of features. $\sigma(\cdot)$ denotes the Sigmoid function to constrain each entry in $\mathbf{S}$ falls into the range of $(0, 1)$. Then we can capture the invariant and complementary spurious features at both structure and attribute granularity in the latent representation space by applying the separating scores to node representations:

$$\mathbf{H}^{\text{Inv}} = \mathbf{H}' \odot \mathbf{S}, \quad \mathbf{H}^{\text{Spu}} = \mathbf{H}' \odot (1 - \mathbf{S}), \tag{9}$$

where $\mathbf{H}^{\text{Inv}}$ and $\mathbf{H}^{\text{Spu}}$ denote the invariant and spurious node representations respectively, and $\odot$ is the element-wise product. Finally, the invariant and spurious representation (denoted as $\mathbf{z}^{\text{Inv}}$ and $\mathbf{z}^{\text{Spu}}$ respectively) of $G$ can be generated by a readout operator:

$$\mathbf{z}^{\text{Inv}} = \text{READOUT}(\mathbf{H}^{\text{Inv}}) \in \mathbb{R}^d, \quad \mathbf{z}^{\text{Spu}} = \text{READOUT}(\mathbf{H}^{\text{Spu}}) \in \mathbb{R}^d. \tag{10}$$

### 4.3 Learning Objective

Our OOD optimization objectives are composed of an invariant learning loss, a task prediction loss, and two additional regularization losses.

**Task-agnostic Self-supervised Invariant Learning.** Invariant learning aims to optimize the encoding $\text{GNN}_E$ and the scoring $\text{GNN}_S$ to produce precise invariant and spurious representations. In particular, we need $\mathbf{z}^{\text{Inv}}$ to be invariant under environmental changes. Additionally, we expect the objective to be independent of the downstream task, which allows the method to be not restricted to a specific type of task. To achieve these, we design a task-agnostic and self-supervised invariant learning objective. Specifically, we disturb $\mathbf{z}_i^{\text{Inv}}$ via concatenating a corresponding $\mathbf{z}_j^{\text{Spu}}$ in a shuffled batch, resulting in an augmented representation $\widetilde{\mathbf{z}}_i^{\text{Inv}}$:

$$\widetilde{\mathbf{z}}_i^{\text{Inv}} = \mathbf{z}_i^{\text{Inv}} \oplus \mathbf{z}_{j \in [1,B]}^{\text{Spu}}, \tag{11}$$

where $\oplus$ denotes concatenation operator and $B$ is batch size.

Inspired by a simple self-supervised learning framework that takes different augmentation views as similar positive pairs and does not require negative samples [37], we treat $\mathbf{z}_i^{\text{Inv}}$ and $\widetilde{\mathbf{z}}_i^{\text{Inv}}$ as positive pairs and push them to be similar, using an MLP-based predictor (denoted as $\omega$) that transforms the output of one view and aligns it to the other view. We minimize their negative cosine similarity:

$$\mathcal{L}_{\text{inv}} = -\sum_{i=1}^{B} \text{sim}(\text{sg}[\mathbf{z}_i^{\text{Inv}}], \omega(\widetilde{\mathbf{z}}_i^{\text{Inv}})), \tag{12}$$

where $\text{sim}(\cdot, \cdot)$ represents the formula of cosine similarity and $\text{sg}[\cdot]$ denotes stop-gradient operation to prevent collapsing [37]. We employ $\mathcal{L}_{\text{inv}}$ as our objective for invariant learning to ensure the invariance of $\mathbf{z}^{\text{Inv}}$ against distribution shifts.

**Task Prediction.** The objective of task prediction is to provide an invariant presentation $\mathbf{z}^{\text{Inv}}$ with sufficient predictive abilities. During training, the choice of prediction loss function depends on the type of task. For classification tasks, we employ the cross-entropy loss, while for regression tasks, we use the mean squared error loss. Take the binary classification task as an example, the cross-entropy loss is computed between the predicted $\widehat{y_i} = \rho(\mathbf{z}_i^{\text{Inv}})$ and the ground-truth label $y_i$:

$$\mathcal{L}_{\text{pred}} = \sum_{i=1}^{B} \left(y_i \log \widehat{y_i} + (1 - y_i) \log\left(1 - \widehat{y_i}\right)\right). \tag{13}$$

Table 1: Evaluation performance on GOOD benchmark. - denotes abnormal results caused by under-fitting declared in the leaderboard, and / denotes that the method cannot be applied to this dataset. The best is marked with **boldface** and the second best is with underline.

| Method | GOOD-HIV ↑ | | | | GOOD-ZINC ↓ | | | | GOOD-PCBA ↑ | | | |
|---|---|---|---|---|---|---|---|---|---|---|---|---|
| | scaffold | | size | | scaffold | | size | | scaffold | | size | |
| | covariate | concept | covariate | concept | covariate | concept | covariate | concept | covariate | concept | covariate | concept |
| ERM | 69.55(2.39) | 72.48(1.26) | 59.19(2.29) | 61.91(2.29) | 0.1802(0.0174) | 0.1301(0.0052) | 0.2319(0.0072) | 0.1325(0.0085) | 17.11(0.34) | 21.93(0.74) | 17.75(0.46) | 15.60(0.55) |
| IRM | 70.17(2.78) | 71.78(1.37) | 59.94(1.59) | -(-) | 0.2164(0.0160) | 0.1339(0.0043) | 0.6984(0.2809) | 0.1336(0.0055) | 16.89(0.29) | 22.37(0.43) | 17.68(0.36) | 15.82(0.52) |
| VREx | 69.34(3.54) | 72.21(1.42) | 58.49(2.28) | 61.21(2.00) | 0.1815(0.0154) | 0.1287(0.0053) | 0.2270(0.0136) | 0.1311(0.0067) | 17.10(0.20) | 21.65(0.82) | 17.80(0.35) | 15.85(0.47) |
| GroupDRO | 68.15(2.84) | 71.48(1.27) | 57.75(2.86) | 59.77(1.95) | 0.1870(0.0128) | 0.1323(0.0041) | 0.2377(0.0147) | 0.1333(0.0064) | 16.55(0.39) | 21.91(0.93) | 16.74(0.60) | 15.21(0.66) |
| Coral | 70.69(2.25) | 72.96(1.06) | 59.39(2.90) | 60.29(2.50) | 0.1769(0.0152) | 0.1303(0.0057) | 0.2292(0.0090) | 0.1261(0.0060) | 17.00(0.38) | 22.00(0.46) | 17.83(0.31) | 16.88(0.58) |
| DANN | 69.43(2.42) | 71.70(0.90) | 62.38(2.65) | 65.15(3.13) | 0.1746(0.0084) | 0.1269(0.0042) | 0.2326(0.0140) | 0.1348(0.0091) | 17.20(0.46) | 22.03(0.72) | 17.71(0.26) | 15.78(0.40) |
| Mixup | 70.65(1.86) | 71.89(1.73) | 59.11(3.11) | 62.80(2.43) | 0.2066(0.0123) | 0.1391(0.0071) | 0.2531(0.0150) | 0.1547(0.0082) | 16.52(0.33) | 20.52(0.50) | 17.42(0.29) | 13.71(0.37) |
| DIR | 68.44(2.51) | 71.40(1.48) | 57.67(3.75) | 74.39(1.45) | 0.3682(0.0639) | 0.2543(0.0454) | 0.4578(0.0412) | 0.3146(0.1225) | 16.33(0.39) | **23.82(0.89)** | 16.04(1.14) | 16.80(1.17) |
| GSAT | 70.07(1.76) | 72.51(0.97) | 60.73(2.39) | 56.96(1.76) | 0.1418(0.0077) | 0.1066(0.0046) | 0.2101(0.0095) | 0.1038(0.0030) | 16.45(0.17) | 20.18(0.74) | 17.57(0.40) | 13.52(0.90) |
| GREA | 71.98(2.87) | 70.76(1.16) | 60.11(1.07) | 60.96(1.55) | 0.1691(0.0159) | 0.1157(0.0084) | 0.2100(0.0081) | 0.1273(0.0044) | 16.28(0.34) | 20.23(1.53) | 17.12(1.01) | 13.82(1.18) |
| CAL | 69.12(1.10) | 72.49(1.05) | 59.34(2.14) | 56.16(4.73) | / | / | / | / | 15.87(0.69) | 18.62(1.21) | 16.92(0.72) | 13.01(0.65) |
| DisC | 58.85(7.26) | 64.82(6.78) | 49.33(3.84) | 74.11(6.69) | / | / | / | / | / | / | / | / |
| MoleOOD | 69.39(3.43) | 69.08(1.35) | 58.63(1.78) | 55.90(4.93) | 0.2752(0.0288) | 0.1996(0.0136) | 0.3468(0.0366) | 0.2275(0.2183) | 12.90(0.86) | 12.92(1.23) | 12.64(0.55) | 10.30(0.36) |
| CIGA | 69.40(1.97) | 71.65(1.33) | 61.81(1.68) | 73.62(1.33) | / | / | / | / | / | / | / | / |
| iMoLD | **72.93(2.29)** | **74.32(1.63)** | **62.86(2.58)** | **77.43(1.32)** | **0.1410(0.0054)** | **0.1014(0.0040)** | **0.1863(0.0139)** | **0.1029(0.0067)** | **17.32(0.31)** | 22.58(0.67) | **18.02(0.73)** | **18.21(1.10)** |

**Scoring GNN Regularization.** For the scoring $\text{GNN}_S$, we apply a penalty to the separation of invariant features, preventing abundance or scarcity and ensuring a reasonable selection. To achieve this, we introduce a regularization term $\mathcal{L}_{\text{reg}}$ to constrain the size of the selected invariant features:

$$\mathcal{L}_{\text{reg}} = \left| \frac{\langle \mathbf{J}, \mathbf{S} \rangle_F}{|\mathcal{V}| \times d} - \gamma \right|, \tag{14}$$

where $\mathbf{J} \in \mathbb{R}^{|\mathcal{V}| \times d}$ denotes a matrix with all entries as 1, $\langle \cdot, \cdot \rangle_F$ is the Frobenius dot product and $\gamma$ is a pre-defined threshold between 0 and 1.

**Codebook Regularization.** In the RVQ module, following VQ-VAE [52, 53], to encourage $\mathbf{h}_v$ to remain close to the selected codebook embedding $\mathbf{e}_k$ in Equation (5) and prevent it from frequently fluctuating between different embeddings, we add a commitment loss $\mathcal{L}_{\text{cmt}}$ to foster that $\mathbf{h}_v$ commits to $\mathbf{e}_k$ and does not grow uncontrollably:

$$\mathcal{L}_{\text{cmt}} = \sum_{v \in \mathcal{V}} \| \text{sg}[\mathbf{e}_k] - \mathbf{h}_v \|_2^2. \tag{15}$$

Finally, the learning objective can be defined as the weighted sum of the above losses:

$$\mathcal{L} = \mathcal{L}_{\text{pred}} + \lambda_1 \mathcal{L}_{\text{inv}} + \lambda_2 \mathcal{L}_{\text{reg}} + \lambda_3 \mathcal{L}_{\text{cmt}}, \tag{16}$$

where $\lambda_1$, $\lambda_2$ and $\lambda_3$ are hyper-parameters to control the weights of $\mathcal{L}_{\text{inv}}$ (in Equation (12)), $\mathcal{L}_{\text{reg}}$ (in Equation (14)) and $\mathcal{L}_{\text{cmt}}$ (in Equation (15)), respectively.

## 5 Experiments

In this section, we conduct extensive experiments to answer the research questions. (**RQ1**) Can our method iMoLD achieve better OOD generalization performance against SOTA baselines? (**RQ2**) How does each of the component we propose contribute to the final performance? (**RQ3**) How can we understand the latent discrete space induced by the proposed RVQ?

### 5.1 Experimental Setup

**Datasets.** We employ two real-world benchmarks for OOD molecular representation learning. Details of datasets are in Appendix A.

- **GOOD** [63], which is a systematic benchmark tailored specifically for graph OOD problems. We utilize three molecular datasets for the graph prediction task: (1) **GOOD-HIV** [16], where the objective is binary classification to predict whether a molecule can inhibit HIV; (2) **GOOD-ZINC** [64], which is a regression dataset aimed at predicting molecular solubility; and (3) **GOOD-PCBA** [16], which includes 128 bioassays and forms 128 binary classification tasks. Each dataset comprises two environment-splitting strategies (scaffold and size), and two shift types (covariate and concept) are applied per splitting outcome, resulting in a total of 12 distinct datasets.

Table 2: Evaluation performance on DrugOOD benchmark. The best is marked with **boldface** and the second best is with underline.

| Method | IC50 ↑ | | | EC50 ↑ | | |
|--------|--------|--------|--------|--------|--------|--------|
| | Assay | Scaffold | Size | Assay | Scaffold | Size |
| ERM | 71.63(0.76) | 68.79(0.47) | 67.50(0.38) | 67.39(2.90) | 64.98(1.29) | 65.10(0.38) |
| IRM | 71.15(0.57) | 67.22(0.62) | 61.58(0.58) | 67.77(2.71) | 63.86(1.36) | 59.19(0.83) |
| Coral | 71.28(0.91) | 68.36(0.61) | 64.53(0.32) | 72.08(2.80) | 64.83(1.64) | 58.47(0.43) |
| MixUp | 71.49(1.08) | 68.59(0.27) | 67.79(0.39) | 67.81(4.06) | 65.77(1.83) | 65.77(0.60) |
| DIR | 69.84(1.41) | 66.33(0.65) | 62.92(1.89) | 65.81(2.93) | 63.76(3.22) | 61.56(4.23) |
| GSAT | 70.59(0.43) | 66.45(0.50) | 66.70(0.37) | 73.82(2.62) | 64.25(0.63) | 62.65(1.79) |
| GREA | 70.23(1.17) | 67.02(0.28) | 66.59(0.56) | 74.17(1.47) | 64.50(0.78) | 62.81(1.54) |
| CAL | 70.09(1.03) | 65.90(1.04) | 66.42(0.50) | 74.54(4.18) | 65.19(0.87) | 61.21(1.76) |
| DisC | 61.40(2.56) | 62.70(2.11) | 61.43(1.06) | 63.71(5.56) | 60.57(2.27) | 57.38(2.48) |
| MoleOOD | 71.62(0.52) | 68.58(1.14) | 65.62(0.77) | 72.69(1.46) | 65.74(1.47) | 65.51(1.24) |
| CIGA | 71.86(1.37) | **69.14(0.70)** | 66.92(0.54) | 69.15(5.79) | 67.32(1.35) | 65.65(0.82) |
| iMoLD | **72.11(0.51)** | 68.84(0.58) | **67.92(0.43)** | **77.48(1.70)** | **67.79(0.88)** | **67.09(0.91)** |

- **DrugOOD** [13], which is a OOD benchmark for AI-aided drug discovery. This benchmark provides three environment-splitting strategies, including assay, scaffold and size, and applies these three splitting to two measurements (IC50 and EC50). As a result, we obtain 6 datasets, and each dataset contains a binary classification task for drug target binding affinity prediction. A detailed description of each environment-splitting strategy is also in Appendix A.

**Baselines.** We thoroughly compare our method against ERM [65] and two groups of OOD baselines: (1) general OOD algorithms used for Euclidean data, which include domain adaptation methods such as Coral [44] and DANN [43], group distributionally robust optimization method (Group-DRO [41]), invariant learning methods such as IRM [19] and VREx [40] and data augmentation method (Mixup [66]). And (2) graph-specific algorithms, including Graph OOD algorithms such as CAL [28], DisC [32], MoleOOD [25] and CIGA [26], as well as interpretable graph learning methods such as DIR [29], GSAT [49] and GREA [47]. Details of baselines and implementation are in Appendix B.

**Evaluation.** We report the ROC-AUC score for GOOD-HIV and DrugOOD datasets as the task is binary classification. For GOOD-ZINC, we use the Mean Average Error (MAE) since the task is regression. While for GOOD-PCBA, we use Average Precision (AP) averaged over all tasks as the evaluation metric due to extremely imbalanced classes. We run experiments 10 times with different random seeds, select models based on the validation performance and report the mean and standard deviations on the test set.

## 5.2 Main Results (RQ1)

Table 1 and Table 2 present the empirical results on GOOD and DrugOOD benchmarks, respectively. Our method iMoLD achieves the best performance on 16 of the 18 datasets and ranks second on the other two datasets. Among the compared baselines, the graph-specific OOD methods perform best on only 11 datasets, and some general OOD methods outperform them on another 7 datasets. This suggests that although some advanced graph-specific OOD methods can achieve superior performance on some synthetic datasets (e.g., to predict whether a specific motif is present in a synthetic graph [29]), they may not perform well on molecules due to the realistic and complex data structures and distribution shifts. In contrast, our method is able to achieve the best performance on most of the datasets, which indicates that the proposed identification of invariant features in the latent space is effective for applying the invariance principle to the molecular structure. We also observe that MoleOOD, a method designed specifically for molecules, does not perform well on GOOD-ZINC and GOOD-PCBA, possibly due to its dependence on inferred environments. Inferring environments may become more challenging for larger-scale datasets, such as GOOD-ZINC and GOOD-PCBA, which contain hundreds of thousands of data and more complex tasks (e.g., PCBA has a total of 128 classification tasks). Our method does not require the inference of environment, and is shown to be effective on datasets of diverse scales and tasks.

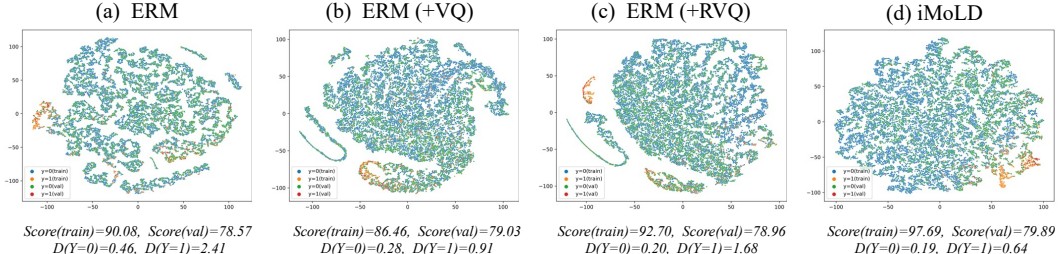

Figure 4: Visualization of the extracted features on training and validation set when the model achieves the highest score on the validation set. *Score(train)* and *Score(val)* are ROC-AUC scores on the training and validation set, respectively. *D(Y=0)* and *D(Y=1)* are distances between the features on the training and validation sets of each class.

## 5.3 Ablation Studies of Components (RQ2)

We conduct ablation studies to analyze how each proposed component contributes to the final performance. Three groups of variants are included: (1) The first group is to investigate the RVQ module. It consists of ERM and ERM (+RVQ), to explore the performance of the simplest ERM and the role of applying the RVQ to it. It also consists of variants of our method, including w/o VQ, which uses $\mathbf{H}$ instead of $\mathbf{H}'$ in Equation (9) to obtain $\mathbf{H}^{\mathrm{Inv}}$ and $\mathbf{H}^{\mathrm{Spu}}$; w/o R, which obtains $\mathbf{H}^{\mathrm{Inv}}$ and $\mathbf{H}^{\mathrm{Spu}}$ without a residual connection by using the output of discretization in Equation (5). (2) The second group is to investigate the impact of each individual training objective, including w/o $\mathcal{L}_{\mathrm{inv}}$, w/o $\mathcal{L}_{\mathrm{reg}}$ and w/o $\mathcal{L}_{\mathrm{cmt}}$, which remove $\mathcal{L}_{\mathrm{inv}}$, $\mathcal{L}_{\mathrm{reg}}$ and $\mathcal{L}_{\mathrm{cmt}}$ from the final objective function in Equation (16), respectively. (3) The third group consists of variants that replace our proposed self-supervised invariant learning objective with other methods, including w/ $\mathcal{L}_{CIGA}$ and w/ $\mathcal{L}_{GREA}$, which represent the objective functions using CIGA [26] and GREA [47], respectively.

Table 3 reports the results on the scaffold-split of GOOD-HIV and the size-split of GOOD-PCBA. We observe that the addition of the RVQ module can improve performance without using any invariant learning algorithms. This indicates that the designed RVQ is effective. Notably, on the GOOD-HIV dataset, removing the VQ module results in significant performance degradation. While on the GOOD-PCBA dataset, removing the residual connection leads to a significant performance drop. This observation may be attributed to the larger scale and more complex task of the GOOD-PCBA dataset. Although VQ im-

Table 3: Ablation studies. / denotes that the variant cannot be applied to this dataset.

| | GOOD-HIV-Scaffold | | GOOD-PCBA-Size | |
|---|---|---|---|---|
| | covariate | concept | covariate | concept |
| iMoLD | **72.93(2.29)** | **74.32(1.63)** | **18.02(0.73)** | **18.21(1.10)** |
| ERM | 69.55(2.39) | 72.48(1.26) | 17.75(0.46) | 15.60(0.55) |
| ERM (+RVQ) | 70.18(1.07) | 73.14(1.99) | 17.84(0.47) | 16.59(0.40) |
| w/o VQ | 70.18(1.60) | 72.88(0.66) | 17.88(1.57) | 17.37(0.82) |
| w/o R | 72.40(2.21) | 73.03(0.15) | 17.37(0.62) | 12.99(2.09) |
| w/o $\mathcal{L}_{\mathrm{inv}}$ | 70.73(1.35) | 71.89(1.80) | 17.55(0.34) | 17.05(0.87) |
| w/o $\mathcal{L}_{\mathrm{reg}}$ | 71.47(0.72) | 72.39(0.87) | 17.33(0.68) | 17.91(0.34) |
| w/o $\mathcal{L}_{\mathrm{cmt}}$ | 70.20(4.07) | 71.98(1.36) | 17.24(1.02) | 17.68(2.20) |
| w/ $\mathcal{L}_{CIGA}$ | 71.23(1.31) | 73.31(1.73) | / | / |
| w/ $\mathcal{L}_{GREA}$ | 72.33(1.23) | 72.94(2.30) | 17.93(0.55) | 16.86(0.50) |

proves the generalization, it reduces the expressivity capacity of the model. Thus, achieving a balance between the benefits and weaknesses brought by VQ is necessary with the proposed RVQ module. By observing the second group, we can conclude that each module is instrumental to the final performance. Removing the invariant learning objective $\mathcal{L}_{\mathrm{inv}}$ and commitment loss $\mathcal{L}_{\mathrm{cmt}}$ results in more pronounced performance degradation, suggesting that the invariant learning objective is effective and indispensable, as well as the need for a constraint term to justify the output of the encoder and the codebook in VQ. By comparing the third group of results, we find that our proposed learning objective outperforms the existing methods, which shows that our approach can not only be applied to a wider range of tasks but also achieve better performance. Moreover, to investigate the robustness of our model, we perform a hyper-parameter sensitivity analysis in Appendix C.1.

## 5.4 Analysis of Latent Discrete Space (RQ3)

To further explore why our method performs better and to understand the superiority of the introduced discrete latent space, we visualize the projection of extracted $\mathbf{z}^{\mathrm{Inv}}$ on both training and validation set

when the model achieves the best score on the validation set, using t-SNE [67] on the covariate-shift dataset of GOOD-HIV-Scaffold in Figure 4 (d). We also visualize the results of some baselines, including the vanilla ERM (Figure 4 (a)), the ERM equipped with VQ after encoder (ERM(+VQ), Figure 4 (b)), and with the RVQ module (ERM(+RVQ), Figure 4 (c)). We additionally compute the 1-order Wasserstein distance [68] between the features on the training and validation sets of each class, to quantify the dissimilarity in the feature distribution across varying environments. We find that adding the VQ or RVQ after the encoder results in a more uniform distribution of features and lower feature distances, due to the fact that VQ makes it possible to reuse previously encountered embeddings in new environments by discretizing them. Moreover, the feature distribution of our method is more uniform and the distance is smaller. This suggests that our method is effective in identifying features that are invariant across different environments. Moreover, we also observe that all ERM methods achieve lower validation scores with higher training scores, implying that these methods are prone to overfitting. Our method, on the other hand, achieves not only a higher validation score but also a higher corresponding training score, thereby demonstrating its ability to overcome the problem of easy overfitting and improve the generalization ability effectively.

## 6 Conclusion

In this work, we propose a new framework that learns invariant molecular representation against distribution shifts. We adopt a "first-encoding-then-separation" strategy, wherein a combination of encoding GNN and residual vector quantization is utilized to derive molecular representation in latent discrete space. Then we learn a scoring GNN to identify invariant features from this representation. Moreover, we design a task-agnostic self-supervised learning objective to enable precise invariance identification and versatile applicability to various tasks. Extensive experiments on real-world datasets demonstrate the superiority of our method on the molecular OOD problem. Overall, our proposed framework presents a promising approach for learning invariant molecular representations and offers valuable insights for addressing distribution shifts in molecular data analysis.

## Acknowledgments

This work was supported by the National Key Research and Development Program of China (2022YFB4500300), National Natural Science Foundation of China (NSFCU19B2027, NSFC91846204, NSFC62302433), joint project DH-2022ZY0012 from Donghai Lab, and sponsored by CCF-Tencent Open Fund (CCF-Tencent RAGR20230122). We want to express gratitude to the anonymous reviewers for their hard work and kind comments.

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

# A  Details of Datasets

## A.1  Dataset

In this paper, we use 18 publicly benchmark datasets, 12 of which are from GOOD [63] benchmark. They are the combination of 3 datasets (GOOD-HIV, GOOD-ZINC and GOOD-PCBA), 2 types of distribution shift (covariate and concept), and 2 environment-splitting strategies (scaffold and size). The rest 6 datasets are from DrugOOD [13] benchmark, including IC50-Assay, IC50-Scaffold, IC50-Size, EC50-Assay, EC50-Scaffold, and EC50-Size. The prefix denotes the measurement and the suffix denotes the environment-splitting strategies. This benchmark exclusively focuses on covariate shift. We use the latest data released on the official webpage[3] based on the ChEMBL 30 database[4]. We use the default dataset split proposed in each benchmark. For covariate shift, the training, validation and testing sets are obtained based on environments without interactions. For concept shift, a screening approach is leveraged to scan and select molecules in the dataset. Statistics of each dataset are in Table 4.

Table 4: Dataset statistics.

| | Dataset | | | Task | Metric | #Train | #Val | #Test | #Tasks |
|---|---|---|---|---|---|---|---|---|---|
| GOOD | HIV | scaffold | covariate | Binary Classification | ROC-AUC | 24682 | 4133 | 4108 | 1 |
| | | | concept | Binary Classification | ROC-AUC | 15209 | 9365 | 10037 | 1 |
| | | size | covariate | Binary Classification | ROC-AUC | 26169 | 4112 | 3961 | 1 |
| | | | concept | Binary Classification | ROC-AUC | 14454 | 3096 | 10525 | 1 |
| | ZINC | scaffold | covariate | Regression | MAE | 149674 | 24945 | 24946 | 1 |
| | | | concept | Regression | MAE | 101867 | 43539 | 60393 | 1 |
| | | size | covariate | Regression | MAE | 161893 | 24945 | 17042 | 1 |
| | | | concept | Regression | MAE | 89418 | 19161 | 70306 | 1 |
| | PCBA | scaffold | covariate | Multi-task Binary Classification | AP | 262764 | 44019 | 43562 | 128 |
| | | | concept | Multi-task Binary Classification | AP | 159158 | 90740 | 119821 | 128 |
| | | size | covariate | Multi-task Binary Classification | AP | 269990 | 43792 | 31925 | 128 |
| | | | concept | Multi-task Binary Classification | AP | 150121 | 32168 | 115205 | 128 |
| DrugOOD | IC50 | assay | | Binary Classification | ROC-AUC | 34953 | 19475 | 19463 | 1 |
| | | scaffold | | Binary Classification | ROC-AUC | 22025 | 19478 | 19480 | 1 |
| | | size | | Binary Classification | ROC-AUC | 37497 | 17987 | 16761 | 1 |
| | EC50 | assay | | Binary Classification | ROC-AUC | 4978 | 2761 | 2725 | 1 |
| | | scaffold | | Binary Classification | ROC-AUC | 2743 | 2723 | 2762 | 1 |
| | | size | | Binary Classification | ROC-AUC | 5189 | 2495 | 2505 | 1 |

## A.2  The Cause of Molecular Distribution Shift

The molecule data can be divided according to different environments, and distribution shifts occur when the source environments of data are different during training and testing. In this work, we investigate three types of environment-splitting strategies, i.e., scaffold, size and assay. And the explanation of each environment are in Table 5.

# B  Details of Implementation

## B.1  Baselines

We adopt the following methods as baselines for comparison, one group of which are common approaches for non-Euclidean data:

- **ERM** [65] minimizes the empirical loss on the training set.

- **IRM** [19] seeks to find data representations across all environments by penalizing feature distributions that have different optimal classifiers.

- **VREx** [40] reduces the risk variances of training environments to achieve both covariate robustness and invariant prediction.

---

[3] https://drugood.github.io/
[4] http://ftp.ebi.ac.uk/pub/databases/chembl/ChEMBLdb/releases/chembl_30

Table 5: Description of different environment splits leading to molecular distribution shifts.

| Environment | Explanation |
| --- | --- |
| Scaffold | Molecular scaffold is the fundamental structure of a molecule with desirable bioactive properties. Molecules with the same scaffold belong to the same environment. Distribution shift arises when there is a change in the molecular scaffold. |
| Size | The size of a molecule refers to the total number of atoms in the molecule. Molecular size is also an inherent structural characteristic of molecular graphs. The distribution shift occurs when size changes. |
| Assay | Assay is an experimental method as an examination or determination for molecular characteristics. Due to variations in assay environments and targets, activity values measured by different assays often differ significantly. Samples tested within the same assay belong to a single environment, while a change in the assay leads to a distribution shift. |

- **GroupDRO** [41] minimizes the loss on the worst-performing group, subject to a constraint that ensures the loss on each group remains close.

- **Coral** [44] encourages feature distributions consistent by penalizing differences in the means and covariances of feature distributions for each domain.

- **DANN** [43] encourages features from different environments indistinguishable by adversarially training a regular classifier and a domain classifier.

- **Mixup** [66] augments data in training through data interpolation.

And the others are graph-specific methods:

- **DIR**[5] [29] discovers the subset of a graph as invariant rationale by conducting interventional data augmentation to create multiple distributions.

- **GSAT**[6] [49] proposes to build an interpretable graph learning method through the attention mechanism and inject stochasticity into the attention to select label-relevant subgraphs.

- **GREA**[7] [47] identifies subgraph structures called rationales by environment replacement to create virtual data points to improve generalizability and interpretability.

- **CAL**[8] [28] proposes a causal attention learning strategy for graph classification to encourage GNNs to exploit causal features while ignoring the shortcut paths.

- **DisC**[9] [32] analyzes the generalization problem of GNNs in a causal view and proposes a disentangling framework for graphs to learn causal and bias substructure.

- **MoleOOD**[10] [25] investigates the OOD problem on molecules and designs an environment inference model and a substructure attention model to learn environment-invariant molecular substructures.

- **CIGA**[11] [26] proposes an information-theoretic objective to extract the desired invariant subgraphs from the lens of causality.

## B.2 Implementation

Experiments are conducted on one 24GB NVIDIA RTX 3090 GPU.

---

[5]https://github.com/Wuyxin/DIR-GNN
[6]https://github.com/Graph-COM/GSAT
[7]https://github.com/liugangcode/GREA
[8]https://github.com/yongduosui/CAL
[9]https://github.com/googlebaba/DisC
[10]https://github.com/yangnianzu0515/MoleOOD
[11]https://github.com/LFhase/CIGA

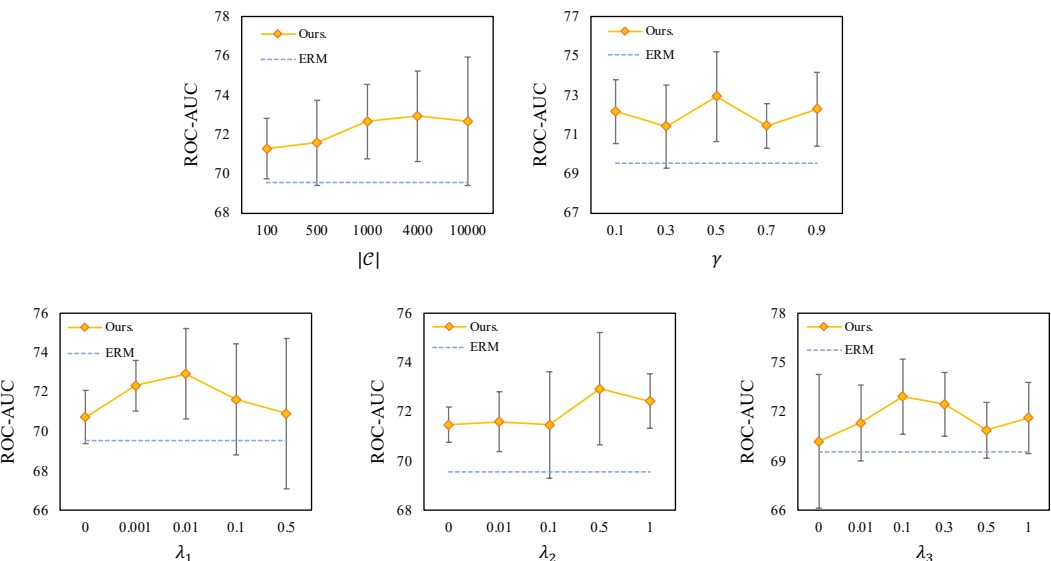

Figure 5: Hyper-parameter sensitivity analysis on the covariate-shift dataset of GOODHIV-Scaffold.

**Baselines.** For datasets in the GOOD benchmark, we use the results provided in the official leaderboard[12]. For datasets in the DrugOOD benchmark, we use the official benchmark code[13] to get the results for ERM, IRM, Coral, and Mixup on the latest version of datasets. The results for GroupDRO and DANN are not reported due to an error occurred while the code was running. For some baselines that do not have reported results, we implement them using public codes. All of the baselines are implemented using the GIN-Virtual [3, 35] (on GOOD) or GIN [35] (on DrugOOD) as the GNN backbone that is parameterized according to the guidance of the respective benchmark. And we conduct a grid search to select hyper-parameters for all implemented baselines.

**Our method.** We implement the proposed iMoLD in Pytorch [69] and PyG [70]. For all the datasets, we select hyper-parameters by ranging the code book size $|\mathcal{C}|$ from $\{100, 500, 1000, 4000, 10000\}$, threshold $\gamma$ from $\{0.1, 0.5, 0.7, 0.9\}$, $\lambda_1$ from $\{0.001, 0.01, 0.1, 0.5\}$, $\lambda_2$ from $\{0.01, 0.1, 0.5, 1\}$, $\lambda_3$ from $\{0.01, 0.1, 0.3, 0.5, 1\}$, and batch size from $\{32, 64, 128, 256, 512\}$. For datasets in DrugOOD, we also select dropout rate from $\{0.1, 0.3, 0.5\}$. The maximum number of epochs is set to 200 and the learning rate is set to 0.001. Please refer to Table 6 for a detailed hyper-parameter configuration of various datasets. The hyper-parameter sensitivity analysis is in Appendix C.1.

## C   Additional Experimental Results

### C.1   Hyper-parameter Sensitivity Analysis

Take the dataset of covariate-shift split of GOODHIV-Scaffold as an example, we conduct extensive experiments to investigate the hyper-parameter sensitivity, and the results are shown in Figure 5. We observe that the performance tends to improve first and then decrease slightly as the size of the codebook $|\mathcal{C}|$ increases. This is because a small codebook limits the expressivity of the model, while too large one cuts the advantage of the discrete space. The effect of the threshold $\gamma$ is insignificant and there is no remarkable trend. As the $\lambda_1$, $\lambda_2$ and $\lambda_3$ increase, the performance shows a tendency to increase first and then decrease, indicating that $\mathcal{L}_{\text{inv}}$, $\mathcal{L}_{\text{reg}}$ and $\mathcal{L}_{\text{cmt}}$ are effective and can improve performance within a reasonable range. We also observe that the standard deviation of performance increases as $\lambda_1$ increases, which may be due to the fact that too much weight on the self-supervised invariant learning objective may enhance or affect the performance. The standard deviation is the smallest when $\lambda_2 = 0$, suggesting that neural networks may have more stable outcomes by learning adaptively when there are no constraints, but it is difficult to obtain higher performance. While the

---

[12]https://good.readthedocs.io/en/latest/leaderboard.html

[13]https://github.com/tencent-ailab/DrugOOD

Table 6: Hyper-parameter configuration.

| | | | $\gamma$ | $|\mathcal{C}|$ | batch-size | $\lambda_1$ | $\lambda_2$ | $\lambda_3$ | dropout |
|---|---|---|---|---|---|---|---|---|---|
| GOOD | HIV | scaffold | covariate | 0.8 | 4000 | 128 | 0.01 | 0.5 | 0.1 | - |
| | | | concept | 0.7 | 4000 | 256 | 0.01 | 0.5 | 0.1 | - |
| | | size | covariate | 0.7 | 4000 | 256 | 0.01 | 0.5 | 0.1 | - |
| | | | concept | 0.9 | 4000 | 1024 | 0.01 | 0.5 | 0.1 | - |
| | ZINC | scaffold | covariate | 0.3 | 4000 | 32 | 0.01 | 0.5 | 0.1 | - |
| | | | concept | 0.5 | 4000 | 256 | 0.01 | 0.5 | 0.1 | - |
| | | size | covariate | 0.3 | 4000 | 256 | 0.01 | 0.5 | 0.1 | - |
| | | | concept | 0.3 | 4000 | 64 | 0.0001 | 0.5 | 0.1 | - |
| | PCBA | scaffold | covariate | 0.9 | 10000 | 32 | 0.0001 | 1 | 0.1 | - |
| | | | concept | 0.9 | 10000 | 32 | 0.0001 | 1 | 0.1 | - |
| | | size | covariate | 0.9 | 10000 | 32 | 0.0001 | 1 | 0.1 | - |
| | | | concept | 0.9 | 10000 | 32 | 0.0001 | 1 | 0.1 | - |
| DrugOOD | IC50 | assay | | 0.7 | 1000 | 128 | 0.001 | 0.5 | 0.1 | 0.5 |
| | | scaffold | | 0.9 | 1000 | 128 | 0.0001 | 0.5 | 0.1 | 0.5 |
| | | size | | 0.7 | 1000 | 128 | 0.01 | 0.5 | 0.1 | 0.1 |
| | EC50 | assay | | 0.7 | 500 | 128 | 0.01 | 0.5 | 0.1 | 0.5 |
| | | scaffold | | 0.3 | 500 | 128 | 0.001 | 0.5 | 0.1 | 0.3 |
| | | size | | 0.7 | 500 | 128 | 0.001 | 0.5 | 0.1 | 0.1 |

standard deviation is the largest when $\lambda_3 = 0$, indicating that the commitment loss in VQ can increase the performance stability.

