# OpenReview forum: "Learning Invariant Molecular Representation in Latent Discrete Space"
_NeurIPS.cc/2023/Conference — NeurIPS 2023 poster_

### Official Review · Reviewer_gGdj · 2023-06-23

**Soundness:** 4 excellent
**Presentation:** 2 fair
**Contribution:** 4 excellent
**Rating:** 6
**Confidence:** 3

**Summary:**

This paper presents a new graph neural network architecture and objective function that encourages models to identify features that are invariant to distribution shifts in the data. The proposed method, iMoLD performs invariant feature extraction in the latent embedding space and leads to improved performance across an extensive set of molecular property prediction tasks.

**Strengths:**

- The presented idea is novel and leads to improved performance across a variety of datasets and tasks.
- Experimentation is extensive with good results.
- The ablation analysis is Section 5.3 and sensitivity analysis from Appendix C are useful.

**Weaknesses:**

#### **Incorrect definitions in Section 3.1**
- I believe there is some issue in the notation of Section 3.1. Specifically, the definitions of $P_{train}, P_{test}, P_{all}$ as collections of distributions, means that they are not themselves valid probability distributions. I believe some re-normalization would be required here.

---

#### **Use of term “Discrete Latent space” is unclear**
- Why do the authors claim they have a **“discrete”** latent space? The residual connection between $\mathbf{H}$ and the quantized representation means that embeddings are continuous. Additionally the element-wise gating to create $\mathbf{H}^{\mathrm{Inv}}$ and $\mathbf{H}^{\mathrm{Spu}}$ means the model does not have a discrete latent representation.

---

#### **Unclear elements about the learning objective**
- The notation in Equation (11) is confusing. Specifically, what is the dimensionality of the $\tilde{\mathbf{z}}_i^{\mathrm{Inv}}$? Are you concatenating multiple batch samples from $\mathbf{z}^{\mathrm{Spu}}$ to $\mathbf{z}_i^{\mathrm{Inv}}$ or just one random one?
- There seems to be an inherent tension between the residual connection and the commitment loss $\mathcal{L}_{\mathrm{cmt}}$. That is, if this loss were perfectly minimized, then the residual connection would be negated.
- The role of $\gamma$ in the scoring regularization is not well described.

---

#### **Baseline presentation is confusing**
- It seems that the authors are conflating baselines in terms of loss objectives and in terms of model/architecture designs. It would be good to clarify which baselines rely on the same architecture but have different objectives (e.g. ERM) and which constitute an entirely different modeling scheme (e.g., CIGA). For the baselines that simply differ in objective, it would be good to also make explicit (could go in Appendix) if any model / architecture adjustments were also applied.

---

#### **Other minor comments**
- In line 147, the notation for edges $\mathcal{E}$ is overloaded, since the same variable is used to denote environments in Section 3.1.
- At the end of Section 5.4 (lines 341-345), the authors seem to be mixing the meaning of low/high in terms of whether low = “good” or low = ”bad”.
- $D$ and $Score$ should be defined explicitly in Figure 4 caption.


**Questions:**

Q1) It is not clear to me why vector quantization (VQ) is the right “bottleneck” to use here. Other than restricting the model’s expressivity, which can be done in other ways such as weight regularization, why is VQ particularly suited for this setup?

Q2) Why is the stop gradient applied in equation 12? Is this simply for computation efficiency / stability? If so, this should be made explicit in the text.

Q3) For the GOOD-PCBA experiment, why is average precision (vs. average accuracy, recall, or ROC-AUC) used?

Q4) I know that there is an extensive sensitivity analysis in the appendix, but what are the hyperparameter configurations for the reported results in the main text (Tables 1 and 2)? Are the “best” iMoLD models sensitive to hyperparameter choice or do you see a general trend as to which configurations perform best?


**Limitations:**

- The current methodology does seem quite intricate with many loss terms that are justified in a somewhat ad hoc manner.
- There is no real discussion of limitations / potential pitfalls relative to previous work.

---

> ### Author Rebuttal · Authors · 2023-08-09
>
> ### (Q1) Distribution definition
> The $P_{train}$, $P_{test}$ and $P_{all}$ given in the manuscript are the form of a collection of distributions, but not a distribution, which needs to be renormalized to avoid ambiguity. We will clarify that in the manuscript.
>
> ### (Q2) The claim of discrete space
> The utilization of the term "discrete space" in the title highlights the incorporation of Vector Quantization (VQ) operation within the methodology. This emphasis is placed to underscore the utilization of VQ in tackling the OOD challenge related to molecules, which is a contribution within this study. We agree that the residual connection between the embedding H and its quantized representation is continuous. Indeed, we did not mean the final representation is discrete. In line 185 we indicate that the final representation is a combination of both continuous and discrete components. We will refine this claim in the final version.
>
> ### (Q3) Some unclear elements
> - In equation (11), the dimensionality of $\widetilde{\mathbf{z}}^{\mathrm{Inv}}_i$ is 2$d$. To derive $\widetilde{\mathbf{z}}^{\mathrm{Inv}}_i$, we first shuffle a batch of $\mathbf{z}^\mathrm{Spu}$ randomly, then concatenate ${\mathbf{z}}^{\mathrm{Inv}}_i$ with the corresponding counterpart $\mathbf{z}^\mathrm{Spu}$ in the shuffled batch.
>
> - The effect of the commitment loss is to restrain the input $\mathbf{h}$ , preventing them from excessively deviating from the embeddings $\mathbf{e}$ within the codebook. It is crucial for $\mathbf{e}$ to be in close proximity to $\mathbf{h}$ to effectively serve as a quantized representation.  When the commitment loss is perfectly minimized, the distribution of $\mathbf{e}$ and $\mathbf{h}$ are overlapped. It is important to note that the residual connection cannot be fully eliminated. This residual gap persists due to inherent limitations imposed by the finite size of the codebook.
>
> - $\gamma$ is used to constrain the size of the selected invariant features. As we employ $\frac{<\mathbf{J},\mathbf{S}>\_{F}}{|\mathcal{V}| \times d} $ to calculate the ratio of identified invariant features. To prevent abundance or scarcity of the invariant features, we set a threshold $\gamma$. This threshold, in turn, functions to optimize the model towards the selection of an invariant feature ratio that closely approximates the predefined value of $\gamma$.
>
> ### (Q4) The baseline presentation
> Thanks for this advice, we will add a table to illustrate the loss objective and model architecture design for each baseline in Appendix.
>
> ### (Q5) Other minor comments
> Thank you for pointing out these issues and we will revise the manuscript correspondingly.
>
> ### (Q6) Why VQ is chosen
> We choose VQ for the following main reasons:
> 1. The citations [56, 57, 58] in lines 122-123 provide theoretical analyses that demonstrate how VQ can enhance noise robustness.
> 2. Intuitively, VQ can act as a bottleneck. In instances where the input is subject to perturbation induced by distribution shifts, the act of discretization emerges as a potent mitigator of such noise, ensuring the output remains unaffected, thereby enhancing modal generalization and alleviating the issue of easy overfitting caused by distribution shifts.
> 3. Certain weight regularization methods, such as IRM, do not perform well when confronted with the molecular OOD problem, as shown in Tables 1 and 2. Their performance is even worse than the ERM baseline.
>
> ### (Q7) The stop-gradient in equation 12
> The stop-gradient operation is used for preventing collapsing. This technique is presented in a self-supervised learning method (citation [36] in line 217). We will clarify this in the manuscript.
>
> ### (Q8) Why average precision is used in PCBA
> Due to the extremely unbalanced classes (only 1.4% positive labels), we use the Average Precision (AP) as the evaluation metric.
>
> ### (Q9) The hyperparameter configuration and the hyperparameter sensitivity
> The chosen hyperparameters are detailed in the uploaded PDF file in the global response. Through parameter sensitivity analysis experiments, we observe that iMoLD outperforms ERM within a reasonable range of parameters. We also observe some general trends through hyperparameter analysis, for example, the performance tends to first improve and then slightly decrease as the codebook size increases, the standard deviation of performance increases as $\lambda_1$ increases, and the standard deviation is the largest when $\lambda_3$ is zero, indicating that the commitment loss in VQ can increase the performance stability.

---

> > ### Comment · Reviewer_gGdj · 2023-08-11
> > **Thank you for the detailed response.**
> >
> > I appreciate the detailed response. I do not have any additional comments or questions at this time.

---

> > > ### Author Response · Authors · 2023-08-22
> > >
> > > We sincerely appreciate your valuable feedback and unwavering support throughout the review process.

---

### Official Review · Reviewer_zQUd · 2023-06-28

**Soundness:** 3 good
**Presentation:** 3 good
**Contribution:** 3 good
**Rating:** 7
**Confidence:** 3

**Summary:**

While significant advances have been made in molecular representation approaches, conventional approaches typically assume that data sources are independent and sampled from the same distribution. However, molecules in real-world drug development often show different characteristics, which might be from a different distribution. This issue is called the out-of-distribution (OOD) problem. OOD challenges the generalization capability of molecular characterization methods and can degrade the performance of downstream tasks. Unlike previous studies' "first-separation-then-encoding" approach, this study proposes a "first-encoding-then-separation" molecular graph representation paradigm. Specifically, the authors first employ a GNN to encode the molecules and then employ a residual vector quantization module to alleviate the overfitting of the training data distribution while preserving the expressiveness of the encoder. Then, they score molecular representations using another GNN that measures the contribution of each dimension to the target in the latent space, thus clearly distinguishing between invariant and spurious representations. Finally, the authors propose a self-supervised learning objective that encourages the recognition of invariant features and effectively preserves label-relevant information while discarding environment-relevant information. The authors conducted experiments on real-world datasets. The experimental results show that the proposed method outperforms the SOTA methods.

**Strengths:**

- Unlike the traditional "first-separation-then-encoding" approach, the authors propose a "first-encoding-then-separation" paradigm that uses an encoding GNN and a scoring GNN to identify invariant features from a graph. The authors use the residual vector quantization module to make a balance between the model's expressivity and generalization. The quantization is used as a bottleneck to strengthen the generalization, and the residual connection complements the model's expressivity. Moreover, the authors design a self-supervised invariant learning objective to facilitate the precise capture of invariant features. This objective is generic, task-independent, and applicable to a variety of tasks.

- The model is cleared described.

- The authors conducted comprehensive experiments on 18 real-world datasets. The experimental results show that the proposed model achieved stronger generalization against SOTA baselines.

- Code has been released and will be valuable for future related research.

**Weaknesses:**

- There are some typos in the paper. For example, a lack of space before references in line 31.

- OOD is repeatedly defined in lines 29 and 90. In addition, please use "OOD" for "out-of-distribution" that appears later in the text, such as lines 130 and 353.

**Questions:**

Could the authors show 3D visualization graphs representing the extracted features, as in Fig. 4?

**Limitations:**

None.

---

> ### Author Rebuttal · Authors · 2023-08-09
>
> ### (Q1) Typos
> Thanks for pointing out these typos, we will revise them accordingly in the updated version.
>
> ### (Q2) 3D visualization
> We provide the 2D and 3D visualization of extracted features together in the uploaded PDF file in the global response. The 3D visualization results show similar characteristics with their 2D counterparts. For instance, the distribution pattern of ERM representations displays increased discreteness, and both ERM(+VQ) and ERM(+RVQ) representations show partially extended flows. In contrast, iMolD yields a more uniform and consistent representation.

---

> > ### Comment · Reviewer_zQUd · 2023-08-22
> >
> > Thanks to authors' effort for improving the paper. No other questions.

---

> > > ### Author Response · Authors · 2023-08-22
> > >
> > > We are truly grateful for your valuable comments and unwavering support.

---

### Official Review · Reviewer_GuEJ · 2023-06-29

**Soundness:** 3 good
**Presentation:** 3 good
**Contribution:** 2 fair
**Rating:** 5
**Confidence:** 2

**Summary:**

This paper proposes a molecular self-supervised learning method for out-of-distribution generalization. The authors introduces a "first-encoding-then-separation" framework to learn invariant features. For doing so, the authors design discrete latent space with VQ-VAE. The experimental results show that their method improves previous baselines in various out-of-distribution downstream tasks.

**Strengths:**

- The paper is well written and easy to understand.

- The pre-training objective to separate invariant and spurious features seems to make sense to me.

**Weaknesses:**

- The complexity of the proposed method is high. The loss function contains several tunable parameters and the ablation study (in Figure 5) shows that the performance is quite dependent on the choice of hyperparameters.

- It seems vague why discrete latent space is needed.

**Questions:**

- How are the hyperparameters chosen in Table 1, 2?

- Is there specific intuition why discrete latent space is useful for out-of-distribution molecular representation learning?

**Limitations:**

Yes, the authors addressed the limitations.

---

> ### Author Rebuttal · Authors · 2023-08-09
>
> ### (Q1) Hyperparameters
> The results presented for the baseline models within the benchmark are derived from an exploration of their respective hyperparameters. In a congruent manner but limited by computational resources, we have searched a partial set of hyperparameters of our method and empirically set some hyperparameters to fixed values. The ranges of the hyperparameters are described in Appendix B.2. And the chosen hyperparameters are detailed in the uploaded PDF file in the global response.
>
> ### (Q2) Intuitive understanding of why discrete latent space
> In this work, we leverage VQ to discretize continuous representations into discrete ones. For every input representation, VQ looks up and fetches the nearest neighbor in a pre-defined codebook and takes it as output. Intuitively, VQ acts as a bottleneck. In instances where the input is subject to perturbation induced by distribution shifts, the act of discretization emerges as a potent mitigator of such noise, ensuring the output remains unaffected. Therefore, the discretization can enhance model generalization and alleviate the easy-over-fitting issue caused by distribution shifts.

---

> > ### Comment · Reviewer_GuEJ · 2023-08-16
> >
> > Thank you for the response. I do not have other comments or questions at this time.

---

> > > ### Author Response · Authors · 2023-08-22
> > >
> > > We sincerely appreciate your positive comments and recognition of the efforts we put into addressing your concerns. Your contribution to our work is highly valued and greatly appreciated.

---

### Official Review · Reviewer_czws · 2023-07-05

**Soundness:** 2 fair
**Presentation:** 2 fair
**Contribution:** 2 fair
**Rating:** 6
**Confidence:** 3

**Summary:**

The paper presents an invariant and robust representation learning approach for molecules to improve the out-of-distribution generalization performance of the predictive models. Specifically, they first map the molecule to the latent representation and then do a separation step where they separate the latent word into invariant and spurious representations.
They also propose using residual vector quantization on the latent representation to avoid over-fitting while preserving the expressiveness power of the encoder.

**Strengths:**

1. The paper tries to address an interesting problem.
2. The proposed idea is novel.
3. They included a detailed ablation study which helps identify the effectiveness of each component.

**Weaknesses:**

1)  The experimental results when compared to the baseline do not have noticeable improvement.
2)  Some more details on the experiment section would be helpful, for example in Figure 4.

**Questions:**

1) Intuitively, what is the difference between applying the Frobenius norm directly to the S matrix versus the regularization defined in equation (14)?

2) It is a bit confusing to use "h" in equation (2) and equation (12) to represent different meanings.

3) It is not clear what effect the discrete term Q(h) has on the learned final note representation H', since H' is the sum of the discrete and continuous representations. Is it possible for the model to completely ignore the discretization step and focus only on the continuous representation?

4) In line 191, it is mentioned that "It is worth noting that our separation is not only performed at the node dimension but also takes into account the feature dimension in the latent space." I didn't quite understand this. Could you provide more explanation?

5) Could you explain the intuition behind what S is learning in equation 8? Essentially, it seems  S is just reweighing every element in H'. Intuitively, for the parts of H' that are not very important/invariant/main motif, S should be low so that those elements mainly contribute to the spurious representation, and vice versa. But how does the model enforces this?

6) I'm not sure if the learned high-level representation can be seen as the sum of the invariant and spurious representations. In other words, can we really break down the abstract learned representation of such a complex structure into invariant and spurious parts? Would each of these components eventually represent some substructures if decoded?

7) The paper states that the model learns discrete latent representation, but according to equation 6, the continuous representation is added back to the discretized representation. Can we still claim that the final learned representation is discrete?

8) It would be very helpful to provide a brief explanation of how the dataset is split, how the out-of-distribution is represented in the training/test/validation data, and what the terms "covariates" and "concepts" refer to in Table 1. This would provide context, especially for readers who are not familiar with the dataset.

**Limitations:**

The paper did not discuss the limitations of the work and there is no potential negative societal impact of the work.

---

> ### Author Rebuttal · Authors · 2023-08-09
>
> ### (Q1) The difference between Frobenius norm and the regularization in Equation (14).
>
> The $<\mathbf{J},\mathbf{S}>\_F$ in Equation (14) is equal to the 1-order Frobenius norm of $S$.
> However, the Frobenius norm merely encapsulates the summation of the elements in the matrix, and the range of its value varies with the size of the matrix. To confine this variability to an appropriate scope, we therefore employ $\frac{<\mathbf{J},\mathbf{S}>\_F}{|\mathcal{V}| \times d} $ to constrain the norm between 0 and 1 despite the variance of matrix size. Intuitively, this is the ratio of the identified invariant features (as delineated in Equation (9) for the invariant and spurious features separation). And to prevent abundance or scarcity of the invariant features, we set a threshold $\gamma$, tasked with optimizing the model to incline towards a selection of invariant attributes' proportion approximating the predefined $\gamma$ value.
>
> ### (Q2) Confusion of notation
> Thank you for pointing this out, we will substitute the "h" in equation (12) with a different notation.
>
> ### (Q3) The effect of the discretization
> VQ discretizes the continuous representation. Intuitively, it plays the role of a bottleneck, constraining the expressive ability of the neural network. This quality of VQ prevents overfitting  on the training distribution and improves the generalization as discussed in lines 119-124. However, VQ also potentially leads to under-fitting because of the limited expressivity. Thus we consider both continuous and discrete representations to strike a balance between generalization and expressivity. And the experimental results in Table 3 verifies the effectiveness of our approach. It can be observed that both discretization and the residual connection contribute to the performance, and removing residual connection leads to a significant performance degradation on PCBA while removing discretization has a more serious impact on HIV.
>
> ### (Q4) Explanation of node dimension and feature dimension
> For the representation matrix $\mathbf{H}^\prime \in \mathbb{R}^{|\mathcal{V}|\times d}$, the node dimension corresponds to the rows, spanning from 1 to $|\mathcal{V}|$, while the feature dimension aligns with the columns, encompassing the range from 1 to $d$. Given that the score matrix $\mathbf{S}$ has the same size with $\mathbf{H}^\prime$, and an element-wise product is conducted with $\mathbf{S}$ and $\mathbf{H}^\prime$ to obtain invariant features. This separation process is employed at each element within the representation matrix, thus encompassing both the node and feature dimensions.
>
> ### (Q5) What S is learning and how to enforce this
> The element at $(i,j)$ position in $\mathbf{S}$ denotes the contribution weight of the $(i,j)$ counterpart in $\mathbf{H}^\prime$ to the invariant representation. For not very important/invariant element in $\mathbf{H}^\prime$, the corresponding element in $\mathbf{S}$ should be low, and vice versa. To enforce the model to produce precise scores as well as invariant representations, we design a task-agnostic self-supervised invariant learning objective as described in Section 4.3, and we use it with task prediction loss to jointly optimize the model.
>
> ### (Q6) Separation into invariant and spurious parts
> Identifying invariant parts is a viable strategy to solve the OOD issue, given that these invariant attributes maintain an exclusive affiliation with labels, remaining unaffected by environment shifts. The remaining parts subsequent to the extraction of these invariant attributes are termed as the "spurious" components. As shown in Figure 1, preliminary studies follow a "first-separation-then-encoding" paradigm, which first divides the graph into invariant and spurious substructures explicitly, and then encode each separately. We argue this practice is suboptimal for extremely complex and entangled molecule graphs. And the detailed motivation is illustrated in the global response. Thus we propose a "first-encoding-then-separation" paradigm, that encodes molecules first and then identify invariant features in the latent space. As mentioned by the reviewer, we cannot ensure complete separation of the invariant and spurious components in the abstract learned representation. This is primarily due to the complex and entangled characteristics of real-world molecule graphs. We can solely rely on analyzing the experimental results to show the enhanced separation achieved, as evidenced by its superior performance compared to other baseline methods. Decoding the invariant and spurious parts in the latent space into structural space is a worthy direction for further investigation, and it is believed that this can provide interpretability to the model. We leave this as future work.
>
> ### (Q7) The claim of discrete representation
> Sorry for the confusion. We did not mean the final representation obtained is discrete. In line 185 we indicate that the final representation is a combination of both continuous and discrete components. We use "discrete space" in the title to emphasize the Vector Quantization (VQ) operations to address molecule OOD problem, which is one of the contributions of this work. We will refine this claim in the final version.
>
> ### (Q8) Explanation of dataset
> We explain "covariate" and "concept" shift in lines 141-145. And we provide details of different distribution shifts in Figure 2 and Appendix A.2. We will provide an explanation of how the dataset is splited in Appendix.

---

### Official Review · Reviewer_d9FY · 2023-07-06

**Soundness:** 3 good
**Presentation:** 3 good
**Contribution:** 2 fair
**Rating:** 5
**Confidence:** 4

**Summary:**

This paper presents a new approach to obtain robust molecular representation through a first-encoding-then-separation method. The proposed method utilizes a graph neural network (GNN) as a molecule encoder and applies a residual vector quantization module to modify the representation. Additionally, a scoring GNN is employed to separate the resulting representations into spurious and invariant categories. The learning process involves contrastive-based self-supervised learning (SSL) loss and task prediction losses. Experimental results on three molecule-related benchmarks demonstrate the superiority of the proposed method over traditional debiasing techniques and recent methods designed specifically for molecule debiasing. Ablation studies and visualization techniques are conducted to provide further insights and analysis.

**Strengths:**

1. The proposed first-encoding-then-separation approach is novel.
2. Experiments on various datasets have shown that the proposed method has the ability to achieve better results.

**Weaknesses:**

1. The motivation is not clear. Why the first-encoding-then-separation approach is reasonable?
2. The reason to combine different components is also not clear, making the technical contribution not strong. The current version is like a straight forward combination without sufficient insight or understanding on the problem. For example, why we need a RVQ module in the molecule representation?
3. It is not clear why the proposed method has the ability to mitigate spurious biases to achieve better OOD results. Is there any theory to support that?
3. The experiments are not sufficient. For example, only improved results have been demonstrated, without sufficient analysis. In the ablation study of different modules are not consistent on different data, making the technique very ad hoc. Though some visualization have been provided, they are not sufficient to support the claim that the proposed method obtain better invariant features. What does it mean by a uniform distribution? Is the uniform distribution equivalent to a good feature?

**Questions:**

See above in the weakness part.

---

> ### Author Rebuttal · Authors · 2023-08-09
>
> ### (Q1) Motivation for "first-encoding-then-separation"
> In contrast to previous works, we adopt a novel paradigm that encodes the whole molecule first, then do separation in the latent representation space. The motivation for this approach is detailed in the global response.
>
> ### (Q2) Technical contribution
> We explain the technical contributions in detail in the global response. Our analysis discerns the unsuitability of dividing in structural space. Instead, we advocate for the "first-encoding-then-separation" paradigm, bolstered by the integration of the RVQ module, which enhances the refinement of the encoded representation. Further, we recognize that the downstream tasks related to molecules are diverse, and some existing methods cannot be applied to all of them. Towards this end, we propose a "task-agnostic self-supervised invariant learning objective" that is applicable to all tasks. We believe our method is devised after we have analyzed the intrinsic differences between molecules and other graph data, with sufficient insight and understanding on the problem.
> ### (Q3) Theory for mitigating spurious biases
> Our approach is theoretically supported. Notably, both the Invariance Principle and the VQ theory lend support to our model, contributing to its enhanced generalization performance.
> 1. _Invariance Principle._ Our method follows the Invariance Principle, which guides us focusing on the causal factors that remain invariant to distribution shifts while overlooking the spurious parts. Formally, our objective can be represented as $\min I({z}^{inv}, {z}^{spu}), \max I({z}^{inv}, y)$，where ${z}^{inv}$ and ${z}^{spu}$ are defined in Equation 10 to represent invariant and spurious features, respectively. To practically achieve $\min I({z}^{inv}, {z}^{spu})$, we leverage $\max I({z}^{inv}, \widetilde{{z}}^{inv})$ as an approximation, as outlined in Equation 12.
> 2. _VQ._ The robustness of noise against distribution shifts gains theoretical validation in citations [56, 57, 58].
>
> [56] Discrete-valued neural communication. NeurIPS'21.
>
> [57] Discrete key-value bottleneck. ICML'23.
>
> [58] Adaptive discrete communication bottlenecks with dynamic vector quantization. AAAI'23.
>
> ### (Q4) Experiments
> We conduct comprehensive experiments in Section 5 and Appendix, including performance comparison on 2 benchmark datasets (Table 1 and Table 2), ablation study (Table 3), visualization (Figure 4) and hyperparameter analysis (Figure 5). We will explain the ablation study and virilization here to address your questions.
>
> ***Ablation study.***
> In the ablation study, we explore 3 groups of model variants. The first two groups involve the sequential removal of individual modules. The third group focuses on the replacement of our proposed task-agnostic self-supervised invariant learning objective, $L_{inv}$ in Equation 16, with alternative counterparts. Specifically, "w/ $L_{CIGA}$" and "$w/ L_{GREA}$" denotes the substitution of $L_{inv}$ with the objective function of CIGA[1] and GREA[2], respectively. The "/" in Table 3 specifies that CIGA's objective function is incompatible with PCBA dataset due to its suitability solely for single-label classification tasks, while PCBA is a multi-label classification task. This is not a case of "different modules are not consistent on different data", but rather highlights the specificity of certain approaches to particular tasks. Our approach, in contrast, boasts a broader applicability that encompasses a range of tasks, setting it apart from task-specific alternatives.
>
> ***Visualization.***
> In visualization, we use two approaches to evaluate the goodness of features:
> - _Distance-based Evaluation._ We measure distances between features in class-specific training and validation sets to assess invariant feature quality, given that features of the same class should be nearer. Illustrated in Fig. 4 titles as "D(Y=0)" and "D(Y=1)", these distances represent data labeled 0 and 1 in training/validation sets. Notably, our method achieves the smallest distance, affirming robustness of our invariant features to environment shifts.
> - _Inference Score Analysis._ We compute inference scores for both training and validation sets, depicted as "Score(train)" and "Score(val)" in Fig. 4 titles. These scores represent performance on respective sets. We observe that other methods could not achieve the highest performance on both two sets, while our method is able to. This demonstrates the ability of our method to overcome the easy overfitting and achieve the best generalization ability.
>
> We employ the above two approaches to assess the quality of features. The uniform distribution that emerges from our model is a noted phenomenon, yet it does not serve as a definitive criterion for evaluating the quality of the features. We use the term “uniform distribution” to refer that the features locate and expand uniformly in the latent space, with few isolated clusters. As depicted in Fig. 4, the visualization outcomes of the ERM, ERM(+VQ) and ERM(+RVQ) exhibit discernible clusters, unlike that obtained through our method.
>
> [1] Learning causally invariant representations for out-of-distribution generalization on graphs. NeurIPS'22.
>
> [2] Graph rationalization with environment-based augmentations. KDD'22.

---

> ### Author Response · Authors · 2023-08-22
>
> We sincerely appreciate your increased rating and recognition of the efforts we put into addressing your concerns. Your contribution to our work is highly valued and greatly appreciated.

---

### Author Rebuttal · Authors · 2023-08-09

We sincerely appreciate the reviewers for providing us with their valuable and affirmative comments regarding our submission.
First of all, we want to clarify our technical contributions. The proposed method mainly consists of three technical contributions:
1. _**The first-encoding-then-separation paradigm.**_ Preliminary studies follow a "first-separation-then-encoding" paradigm, where the graph is first divided into invariant and spurious substructures, and then each part is encoded separately. We argue that this paradigm is suboptimal for extremely complex and entangled molecules, as we have illustrated in lines 48-53, since molecular intricate properties usually cannot be easily determined by analyzing a subset of molecular structures. Citations [29,30] in line 53, which discuss the necessity and importance of considering the molecule as a whole in the study of molecular physicochemical properties, substantiate the motivation. To go into more detail, molecules are composed of atoms, and atoms contact through a cloud of electrons to form covalent bonds. Typically, chemical properties are expressed through electrons, which can conduct across the backbone. The surface potential, polarity, and other chemical properties on some substructures are affected by the charges of atoms in other substructures. The explicit separation of the molecular structure entails a loss of information to inter-substructural interactions. Therefore, distinguishing between invariant and spurious parts in the structural space is suboptimal for molecules and we propose a "first-encoding-then-separation" paradigm to make the distinction in the latent representation space.
2. _**The RVQ module.**_ Since we do not divide the molecule in the structural space, but encode the whole molecule directly. When the environment changes, the obtained representation will be disturbed by the distribution shift noise. To alleviate this issue, we propose to use VQ to improve the generalization ability by discretizing the continuous representation. However, we observe that VQ would also limit the model's expressivity and potentially lead to under-fitting. To address this concern, we propose to equip VQ with a residual connection to strike a balance between generalization and expressivity.
3. _**The task-agnostic self-supervised invariant learning objective.**_ Following the Invariance Principle, we investigate to focus on the causal factors that remain invariant to distribution shifts while overlooking the spurious parts. To accurately make a separation between invariant features and spurious features, an invariant learning objective is needed. We have analyzed that downstream tasks related to molecules are diverse, including regression, single- and multi-label classification. However, the existing invariant learning objectives cannot be applied to certain molecular tasks/datasets (marked with "/" in Table 1). Therefore, we propose the task-agnostic self-supervised invariant learning objective, which is independent of the downstream task and makes our method applicable to various tasks.

In the following sections, we present a detailed point-by-point response to the questions raised.

---

### Decision · Program_Chairs · 2023-09-21

**Decision:**

Accept (poster)

**Comment:**

This paper proposes a method to learn invariant molecule representation using a "first encoding and then separation" scheme. The proposed method employs two graph neural networks. The first one embeds the molecules with residual vector quantization, and the second one scores the nodes and features. The invariant representation is learned by a contrastive self-supervised learning loss, together with classification loss, and regularization. Experiments on benchmarks demonstrate the effectiveness of the proposed method as compared to debiasing methods.

The paper has been reviewed by 5 reviewers. All the reviews are positive after rebuttal and discussion.

Initially, the reviewers raised some concerns, such as the role of discrete latent space, hyper-parameter configuration, motivation, and technical contributions. These concerns were adequately addressed by the authors. One reviewer raised the rating to 5.

Overall, this paper makes a useful contribution to invariant molecule representation.